# (De)Randomized Smoothing for Certifiable Defense against Patch Attacks

**Alexander Levine and Soheil Feizi**
Department of Computer Science
University of Maryland
College Park, MD 20742
{alevine0, sfeizi}@cs.umd.edu

## Abstract

Patch adversarial attacks on images, in which the attacker can distort pixels within a region of bounded size, are an important threat model since they provide a quantitative model for physical adversarial attacks. In this paper, we introduce a certifiable defense against patch attacks that guarantees for a given image and patch attack size, no patch adversarial examples exist. Our method is related to the broad class of *randomized smoothing* robustness schemes which provide high-confidence probabilistic robustness certificates. By exploiting the fact that patch attacks are more constrained than general sparse attacks, we derive meaningfully large robustness certificates against them. Additionally, in contrast to smoothing-based defenses against $L_p$ and sparse attacks, our defense method against patch attacks is de-randomized, yielding improved, deterministic certificates. Compared to the existing patch certification method proposed by Chiang et al. (2020), which relies on interval bound propagation, our method can be trained significantly faster, achieves high clean and certified robust accuracy on CIFAR-10, and provides certificates at ImageNet scale. For example, for a $5 \times 5$ patch attack on CIFAR-10, our method achieves up to around $57.6\%$ certified accuracy (with a classifier with around $83.8\%$ clean accuracy), compared to at most $30.3\%$ certified accuracy for the existing method (with a classifier with around $47.8\%$ clean accuracy). Our results effectively establish a new state-of-the-art of certifiable defense against patch attacks on CIFAR-10 and ImageNet.

## 1 Introduction

In recent years, adversarial attacks have become a topic of great interest in machine learning [1, 2, 3]. However, in many instances the threat models considered for these attacks (e.g. small $L_\infty$ distortions to every pixel of an image) implicitly require the attacker to be able to directly interfere with the input to a neural network. This limits practicality of such attacks as well as defenses against them. On the other hand, the development of *physical* adversarial attacks [4], in which small visible changes are made to real world objects in order to disrupt classification of images of these objects, represents a more concerning security threat. Unlike $L_p$ attacks, physical adversarial attacks can be perceptible (e.g. adding an adversarial sticker on a stop sign is a perceptible change). Nevertheless, humans would still correctly classify the attacked image while the classification model would fail to predict the correct label. Therefore, the attacked image is an adversarial example.

Physical adversarial attacks can often be modeled as "patch" adversarial attacks, in which the attacker can make arbitrary changes to pixels within a region of bounded size. Indeed, there is often a direct relationship between the two: for example, the universal patch attack proposed by [5] is an effective physical sticker attack. The attack method proposed in [5] is universal in a sense that pixels of the

adversarial patch do not depend on the attacked image. Image-specific patch attacks have also been proposed, such as LaVAN [6], which reduces ImageNet classification accuracy to 0% using only a $42 \times 42$ pixel square patch (on images of size $299 \times 299$). In this paper, we consider *all* attacks (image-specific or universal) on square patches of size $m \times m$.

Practical defenses against patch attacks have been proposed.[7, 8] For the aforementioned $42 \times 42$ pixel attacks on ImageNet, [8] claims the current state-of-the-art practical defense. However, [9] has recently broken this defense, reducing the classification accuracy on ImageNet to 14%. In the same work, [9] also proposes the first *certified* defense against patch adversarial attacks, which uses interval bound propagation [10]. In a certifiably robust classification scheme, in addition to providing a classification for each image, the classifier may also return an assurance that the classification will provably not change under any distortion of a certain magnitude and threat model. One then reports both the *clean accuracy* (normal accuracy) of the model, as well as the *certified accuracy* (percent of images which are both correctly classified, and for which it is guaranteed that the classification will not change under a certain attack type). Unlike practical defenses, certified defenses guarantee that *no* future adversary (under a certain threat model) will break the defense.

The certified defense proposed by [9], however, does not scale well to practical classification tasks on complex inputs such as CIFAR-10 or ImageNet samples. Specifically, while this certified defense performs well on MNIST, it achieves poor certified accuracy on CIFAR-10 and, to quote from the paper itself, "is unlikely to scale to ImageNet." In this work, we propose a certified defense against patch attacks which overcomes these issues. In particular, our certifiable defense method leads to the following results:

| Dataset and Attack Size | Chiang et al. [9] Certified Acc (Clean Acc) | Our method Certified Acc (Clean Acc) |
|---|---|---|
| MNIST $5 \times 5$ | **60.4% (92.0%)** | 52.44% (96.54%) |
| CIFAR $5 \times 5$ | 30.3% (47.8%) | **57.58% (83.82%)** |
| ImageNet $42 \times 42$ | N/A | **13.9% (44.6%)** |

Table 1: Comparison of the certified accuracy of our defense vs. [9]. For each technique, we report the certified and clean accuracies of the model with parameters giving the highest certified accuracy.

Notably, our method achieves a more than 27 percentage point increase in certified robustness on CIFAR-10 compared to [9]. Moreover, our method has top-1 *certified* accuracy on ImageNet classification which is approximately equal to the 14% *empirical* accuracy of the state-of-the art practical defense [8] under the attack proposed by [9] (although our clean accuracy is lower, 44% vs. 71%). On MNIST, which is often regarded as a toy dataset in deep learning applications, our method also achieves a relatively high certified robustness (but not as high as the method of [9]) and clean accuracy (slightly higher than that of [9]). Further, the certified defense proposed by [9] also has a computationally expensive training algorithm: the training time for the reported best model was 8.4 GPU hours for MNIST, and 15.4 GPU hours for CIFAR-10, using NVIDIA 2080 Ti GPUs. Our models, by contrast, took approximately 1.0 GPU hour to train on MNIST, and 2.5 GPU hours to train on CIFAR-10, on the same model of GPU.

Our certifiably robust classification scheme is based on *randomized smoothing*, a class of certifiably robust classifiers which have been proposed for various threat models, including $L_2$ [11, 12, 13], $L_1$ [14] and $L_0$ [15, 16] and Wasserstein [17] metrics. All of these methods rely on a similar mechanism where noisy versions of an input image $\mathbf{x}$ are used in the classification. Such noisy inputs are created either by adding random noise to all pixels [14] or by removing (*ablating*) some of the pixels [16]. A large number of noisy images are then classified by a *base classifier* and then the consensus of these classifications is reported as the final classification result. For an adversarial image $\mathbf{x}'$ at a bounded distance from $\mathbf{x}$, the probability distributions of possible noisy images which can be produced from $\mathbf{x}$ and $\mathbf{x}'$ will substantially overlap. This implies that, if a sufficiently large fraction of noisy images derived from $\mathbf{x}$ are classified to some class $c$, then with high confidence, a plurality of noisy images derived from $\mathbf{x}'$ will also be assigned to this class.

Patch adversarial attacks can be considered a special case of $L_0$ (sparse) adversarial attacks: in an $L_0$ attack, the adversary can choose a limited number of pixels and apply unbounded distortions to them. A patch adversarial attack is therefore a sparse adversarial attack where the attacker is additionally

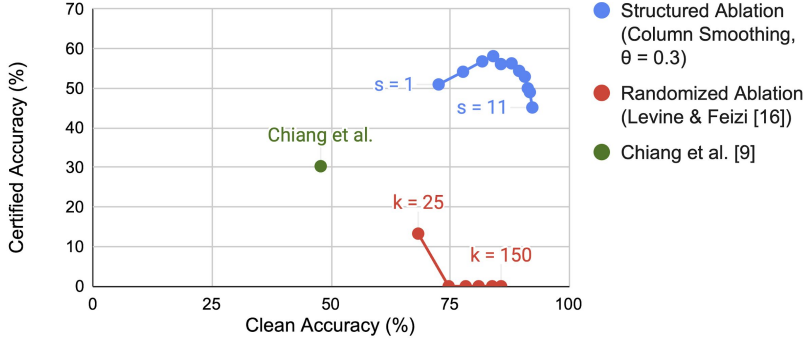

Figure 1: Clean and Certified accuracies for $5 \times 5$ adversarial patches on CIFAR-10. We compare our proposed method, Structured Ablation (for a range of its hyperparameter $s$) with the certified defense for patch attacks proposed by [9], and with a naive application of the $L_0$ defense proposed by [16] (for a range of that technique's hyperparameter $k$). Our defense achieves significantly higher certified and clean accuracies compared to the other methods.

constrained to selecting only a block of adjacent pixels to attack, rather than any arbitrary pixels. The current state-of-the-art certified defense against sparse adversarial attacks is a randomized smoothing method proposed by [16]. In this method, a base classifier, $f(\mathbf{x})$, is trained to make classifications based on only a small number of independently randomly-selected pixels: the rest of the image is *ablated*, meaning that it is encoded as a null value. At test time, the final classification $g(\mathbf{x})$ is taken as the class most likely to be returned by $f$ on a randomly ablated version of the image. In practice, we find that applying the defense method developed in [16] for sparse attacks directly to patch attacks yields poor results (see Figure 1). This is because the defense proposed in [16] does not incorporate the additional structure of the attack. For patch attacks, we can use the fact that the attacked pixels form a contiguous square to develop a more effective defense. In this paper, we propose a *structured ablation* scheme, where instead of independently selecting pixels to use for classification, we select pixels in a correlated way in order to reduce the probability that the adversarial patch is sampled. Empirically, structured ablation certificates yields much improved certified accuracy to patch attacks, compared to the naive $L_0$ certificate.

By reducing the total number of possible ablations of an image, structured ablation allows us to de-randomize our algorithm, yielding improved, deterministic certificates. For $L_0$ robustness, [16] achieves the largest median certificates on MNIST by using a base classifier $f$ which classifies using only 45 out of 784 pixels. There are $\binom{784}{45} \approx 4 \times 10^{73}$ ways to make this selection. It is therefore not feasible to evaluate precisely the probability that $f(\mathbf{x})$ returns any particular class $c$: one must estimate this based on random samples. Using our proposed methods, the number of possible ablations is small enough so that it is tractable to classify using all possible ablations: we can *exactly* evaluate the probability that $f(\mathbf{x})$ returns each class. Our certificate is therefore exact, rather than probabilistic, so our classifications are provably robust in an absolute sense.

Determinism provides another benefit: the absence of estimation error increases the certified accuracies that can be reported. Additionally, because estimation error is no longer a concern, derandomization allows us to use more rich information from the base classifier without incurring an additional cost in increased estimation error. We take advantage of this to allow the base classifier to abstain in cases where it cannot make a high-confidence prediction towards any class. This leads to substantially increased certificates on MNIST, although the effects on CIFAR-10 are not significant.

After the initial distribution of this work, [18] improved upon it by proposing a tighter certificate. In a concurrent work to ours, [19] also proposes a method similar to "block smoothing" proposed below.

## 2   Certifiable Defenses against Patch Attacks

### 2.1   Baseline: Sparse Randomized Ablation [16]

As mentioned in the introduction, patch attacks can be regarded as a restricted case of $L_0$ attacks. In particular, let $\rho$ be the magnitude of an $L_0$ adversarial attack: the attacker modifies $\rho$ pixels and

leaves the rest unchanged. A patch attack, with an $m \times m$ adversarial patch, is also an $L_0$ attack, with $\rho = m^2$. We can then attempt to apply existing certifiably robust classification schemes for the $L_0$ threat model to the patch attack threat model: we simply need to certify to an $L_0$ radius of $\rho = m^2$. Consider specifically the $L_0$ smoothing-based certifiably robust classifier introduced by [16]. In this classification scheme, given an input image $\mathbf{x}$, the base classifier $f$ classifies a large number of distinct randomly-ablated versions of $\mathbf{x}$, in each of which only $k$ pixels of the original image are randomly and independently selected to be retained and used by the base classifier $f$. Therefore, for *any choice* of $\rho$ pixels that the attacker could choose to attack, the probability that any of these $\rho$ pixels is also one of the $k$ pixels used in $f$'s classification is:

$$\Delta := \Pr(f \text{ uses attacked pixels})$$

$$= 1 - \frac{\binom{hw-\rho}{k}}{\binom{hw}{k}} \approx k\frac{\rho}{hw} = \frac{km^2}{hw} \quad (k, \rho << hw),$$

where $\rho$ is the number of attacked pixels, $k$ is the number of retained pixels used by the base classifier, and the overall dimensions of the input image $\mathbf{x}$ are $h \times w$. To understand this, note that the classifier has $k$ opportunities to choose an attacked pixel, and $\rho$ out of $hw$ pixels are attacked. Clearly, if $f$ does not use any of the attacked pixels, then its output will not be corrupted by the attacker. Therefore, the attacker can change the output of $f(\mathbf{x})$ with probability at most $\Delta$. Let $c$ be the majority classification at $\mathbf{x}$ (i.e., $g(\mathbf{x}) = c$). If $f(\mathbf{x}) = c$ with probability greater than $0.5 + \Delta$, then for any distorted image $\mathbf{x}'$, one can conclude that $f(\mathbf{x}') = c$ with probability greater than $0.5$, and therefore that $g(\mathbf{x}') = c$. As discussed in the introduction, while this technique produces state-of-the-art guarantees against general $L_0$ attacks, it yields rather poor certified accuracies when applied to patch attacks, because it does not take advantage of the structure of the attack (See Figure 1; data for MNIST are provided in supplementary material.).

## 2.2 Proposed Method: Structured Ablation

To exploit the restricted nature of patch attacks, we propose two *structured ablation* methods, which select correlated groups of pixels to reduce the probability $\Delta$ that the adversarial patch is sampled:

- **Block Smoothing**: In this method, we select a single $s \times s$ square block of pixels, and ablate the rest of the image. The number of retained pixels is then $k = s^2$. Note that for an $m \times m$ adversarial patch, out of the $h \times w$ possible selections for blocks to use for classification, $(m + s - 1)^2$ of them will intersect the patch. Thus, we have:

$$\Delta_{\text{block}} = \frac{(m + s - 1)^2}{hw} = \frac{(m + \sqrt{k} - 1)^2}{hw} < \frac{4\max(m^2, k)}{hw}. \tag{1}$$

As illustrated in Figure 2, this implies a substantially decreased probability of intersecting the adversarial patch, compared to sampling $k$ pixels independently.

- **Band Smoothing**: In this method, we select a single band (a **column** or a **row**) of pixels of width $s$, and ablate the rest of the image. In the case of a column, the number of retained pixels is then $k = sh$. For an $m \times m$ adversarial patch, out of the $w$ possible selections for bands to use for classification, $m + s - 1$ of them will intersect the patch. Then we have:

$$\Delta_{\text{col.}} = \frac{m + s - 1}{w} = \frac{m + k/h - 1}{w} < \frac{2\max(hm, k)}{hw}. \tag{2}$$

For both of these methods, it is tractable to use the base classifier to classify all possible ablated versions of an image (i.e. $hw$ and $w$ possible ablations for block and column smoothing, respectively). This allows us to exactly compute the smoothed classifier, $g(\mathbf{x})$, yielding deterministic certificates.

Our experiments show that structured ablation produces higher certified accuracy than $L_0$ randomized ablation. This is because, for similar values of $\Delta$, structured ablation methods yield much higher base classifier accuracies (Figure 3). Empirically, we find that the band method (and specifically, column smoothing) produces the most certifiably robust classifiers (Figure 5). In supplementary materials, we explore structured ablation using multiple blocks or bands of pixels.

We now explicitly describe our algorithms, starting with block smoothing. For an input image $\mathbf{x}$, let the base classifier be specified as $f_c(\mathbf{x}, s, x, y)$, where $\mathbf{x}$ is the input image, $s$ is the block size,

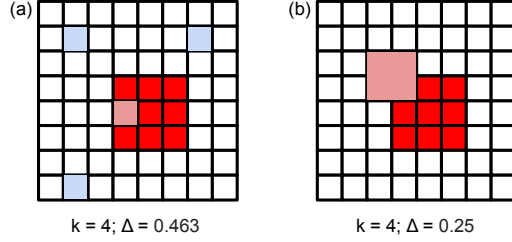

Figure 2: Likelihood of selecting a pixel which is part of the attacked patch (red) for (a) sparse randomized ablation, as proposed by [16] (b) Structured ablation, using a block of size $s = 2$. In both cases, $k = 4$ pixels are retained. However, in the sparse case, if *any* of the four independently-selected pixels sample the patch, then the classification may be impacted: this occurs with probability $\Delta = 1 - \binom{64-9}{4}/\binom{64}{4} \approx 0.463$. In contrast, the probability that the block overlaps with the adversarial patch is only $\frac{16}{64} = 0.25$.

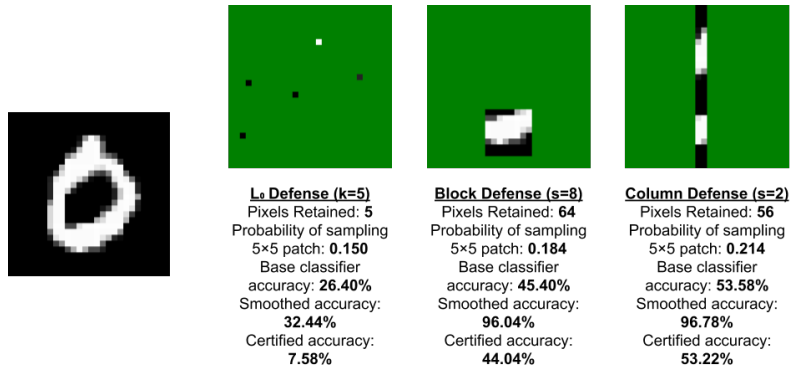

Figure 3: Comparison of the $L_0$ defense proposed by [16] to our proposed defenses on MNIST, for a $5 \times 5$ patch attack. While sampling a single block or band slightly increases the probability $\Delta$ that an adversarially distorted pixel is used, the large increase in the total number of retained pixels, and therefore the base classifier accuracy, more than makes up for this increase in $\Delta$. However, the number of retained pixels alone does not perfectly correspond to higher base classifier accuracy: while the band method uses slightly fewer pixels than the block method, the base classifier has substantially higher accuracy, leading to higher certified accuracy.

$(x, y)$ is the position of the retained block, and $c \in \mathbb{N}$ is a class label. In other words, $f(\mathbf{x}, s, x, y)$ is the base classification, where the classifier uses only the pixels in an $s \times s$ block with upper-left corner $(x, y)$ (If the retained block would exceed the borders of the image, it wraps around: see Figure 4). For each class $c$, $f_c(\mathbf{x}, s, x, y)$ will either be 0 or 1; however, note that we *do not* require that $f_c(\mathbf{x}, s, x, y) = 1$ for any class $c$ (it may abstain, returning zero for all classes), and we also allow for $f_c(\mathbf{x}, s, x, y)$ to equal 1 for multiple classes (see Section 2.2.1 for details). To make our final classification and compute our robustness certificate, we count the number of blocks on which the base classifier returns each class:

$$\forall c, \quad n_c(\mathbf{x}) := \sum_{x=1}^{w} \sum_{y=1}^{h} f_c(\mathbf{x}, s, x, y) \tag{3}$$

The final smoothed classification is simply the plurality class returned: $g(\mathbf{x}) := \arg\max_c n_c(\mathbf{x})$. In the case of ties, we deterministically return the smaller-indexed class. Because the adversarial patch only intersects $(m + s - 1)^2$ blocks, the adversary can only alter the output of $(m + s - 1)^2$ of the evaluations of the base classifier. This yields the following guarantee:

**Theorem 1.** *For any image* $\mathbf{x}$*, base classifier* $f$*, smoothing block size* $s$*, and patch size* $m$*, if:*

$$n_c(\mathbf{x}) \geq \max_{c' \neq c} \left[ n_{c'}(\mathbf{x}) + \mathbf{1}_{c > c'} \right] + 2(m + s - 1)^2 \tag{4}$$

*then for any image* $\mathbf{x}'$ *which differs from* $\mathbf{x}$ *only in an* $(m \times m)$ *patch,* $g(\mathbf{x}') = c$.

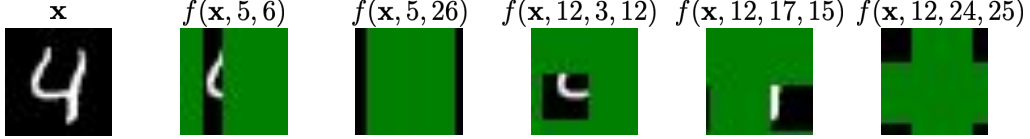

$$\mathbf{x} \qquad f(\mathbf{x}, 5, 6) \qquad f(\mathbf{x}, 5, 26) \qquad f(\mathbf{x}, 12, 3, 12) \quad f(\mathbf{x}, 12, 17, 15) \quad f(\mathbf{x}, 12, 24, 25)$$

Figure 4: Representation of which pixels are used by the base classifier $f$, as a function of indexing. Ablated pixels are represented in green.

In Theorem 1, the indicator function term ($\mathbf{1}_{c>c'}$) is present because we break ties deterministically by label index during the final classification. Proofs are provided in supplementary materials.

Note that the classifier counts $n_c(\mathbf{x})$ can be though of as exact estimates for the probability that the base classifier returns the class $c$, simply scaled up by a factor of $hw$. For the column smoothing case (or row smoothing, by simple transpose), we can compute a similar certificate. In this case, the base classifier is $f_c(\mathbf{x}, s, x)$, where $s$ is now the width of the retained column of pixels, and $x$ is the position of the leftmost edge of this column. We then only need to sum over one dimension:

$$\forall c, \;\; n_c(\mathbf{x}) := \sum_{x=1}^{w} f_c(\mathbf{x}, s, x). \tag{5}$$

Again, we classify using $g(\mathbf{x}) := \arg\max_c n_c(\mathbf{x})$. To derive the final guarantee, we now use that the adversarial patch will overlap with only $(m + s - 1)$ columns:

**Theorem 2.** *For any image* $\mathbf{x}$*, base classifier* $f$*, smoothing column size* $s$*, and patch size* $m$*, if:*

$$n_c(\mathbf{x}) \geq \max_{c' \neq c} \left[ n_{c'}(\mathbf{x}) + \mathbf{1}_{c>c'} \right] + 2(m + s - 1) \tag{6}$$

*then for any image* $\mathbf{x}'$ *which differs from* $\mathbf{x}$ *only in an* $(m \times m)$ *patch,* $g(\mathbf{x}') = c$.

### 2.2.1 Implementation Details

In practice, we use a deep network as our base classifier, and set $f_c(\mathbf{x}, s, x, y) = 1$ if the logit corresponding to class $c$ is greater than a threshold hyperparameter $\theta$. This allows the base classifier to abstain from classifying in the case that there is no usable information in the retained block, as well as to "vote" for multiple classes, which may be beneficial if the base classifier top-1 accuracy is low.

The input of to the neural network used as the base classifier is a copy of the image $\mathbf{x}$, with all pixels except for those in the retained block or band replaced with a specially-encoded 'NULL' value. We encode the additional 'NULL' value in the input in the same manner described for randomized ablation by [16] for each dataset tested: this involves adding additional color channels, so that the NULL value is distinct from all real pixel colors. During training, as in prior smoothing works, we train $f$ on ablated samples, using a single randomly-determined ablation pattern (selection of block or column to retain) on all samples in each batch.

### 2.3 Comparison to Conventional Randomized Smoothing

In conventional randomized smoothing, rather than computing the probability that $f$ returns each class directly, one must instead lower-bound, with high confidence, the probability $p_c$ that $f$ returns the plurality class $c$ and upper-bound the probabilities $p_{c'}$ that $f$ returns all other classes, based on samples. This leads to decreased certified accuracy due to estimation error. Additionally, all of these bounds must hold simultaneously: in order to ensure that the gap between $p_c$ and $p_{c'}$ is sufficiently large for each $c'$ to prove robustness, one must bound the population probabilities for *every* class. Some works [14, 20] do this directly using a union bound, leading to increased error as the number of classes increases. Others, following [12], instead only use samples to lower-bound the probability $p_c$ that the base classifier returns the top class. One can then upper bound all other class probabilities by observing that $\forall c', \; p_{c'} \leq 1 - p_c$. In other words, rather than determining whether $c$ will stay the plurality class at an adversarial point, one instead determines whether $c$ will stay the *majority* class. This is also the estimation method used by [16] for $L_0$ certificates: this is why, when describing that method in Section 2.1, we gave the condition for certification as $p_c > 0.5 + \Delta$. In our deterministic method, we can use a less strict condition, that $\forall c', \; p_c - p_{c'} > 2\Delta$, where $p_c = n_c/hw$ for block

smoothing, and $p_c = n_c/w$ for column smoothing. (As described above, we can sometimes even certify in the *equality* case, when it is assured that $c$ will be selected if there is a tie between the class probabilities at the distorted point.)

In this work, we sidestep the estimation problem entirely by computing the population probabilities exactly. This substantially reduces evaluation time: for example, column smoothing on CIFAR-10 requires 32 forward passes, compared to $10^4 - 10^5$ for randomized ablation [16]. (We provide measured evaluation times in supplementary material.) However, by avoiding the assumption of [12], that all probability not assigned to $c$ is instead assigned to a single adversarial class, we can make an additional optimization: we can add an 'abstain' option. If there is no compelling evidence for any particular class in an ablated image (i.e., if all logits are below a threshold value $\theta$), our classifier abstains. This prevents blocks which contain no information from being assigned to an arbitrary, likely incorrect class. Figure 5-a shows that this significantly increases the certified accuracy on MNIST, although it has little effect on CIFAR-10. Our threshold system also allows the base classifier to select *multiple* classes, if there is strong evidence for each of them. This is intended to increase certified accuracy in the case of a large number of classes, where the top-1 accuracy of the base classifier might be low: if the correct class consistently occurs within the top several classes, it may still be possible to certify robustness.

In a concurrent work, [21] also proposes a derandomization of a randomized smoothing technique. However, the threat model considered is quite different: [21] develops a defense against label-flipping poisoning attacks, where the adversary changes the labels of training samples. Notably, [21]'s result only applies directly to linear base classifiers. By making this restriction, [21] is able to analytically determine the probabilities of $f(\mathbf{x})$ returning each class. By contrast, our de-randomized technique for patch attacks does not restrict the architecture of the base classifier $f$, in practice a deep network.

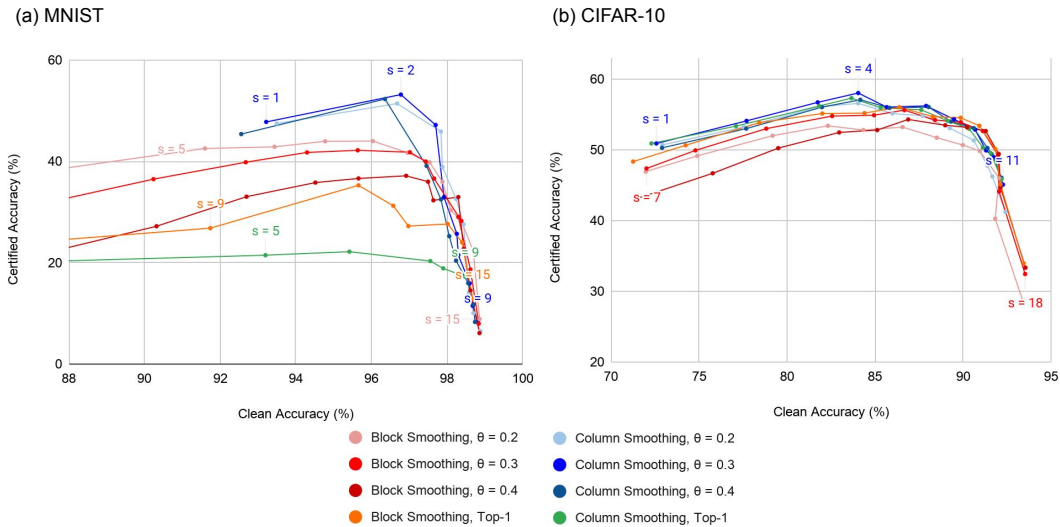

Figure 5: Validation set certificates for $5 \times 5$ patches on (a) MNIST, (b) CIFAR-10. Best certified accuracy is achieved using Column Smoothing for both datasets, with $s = 2, \theta = 0.3$ for MNIST and $s = 4, \theta = 0.3$ for CIFAR-10. Column smoothing (blue lines) gives better certified accuracies than block smoothing (red lines), but the effect is small on CIFAR-10.

## 3 Results

Certified robustness against patch attacks is presented for $5 \times 5$ patches on MNIST and CIFAR-10 in Figure 5, using both block and column smoothing (On MNIST, we also tested smoothing with rows rather than columns, with slightly worse results: see supplementary materials.) Results in the figures are using a validation set of 5,000 images; the final results reported in Table 1 are on a separate test set of 5,000 images. On both datasets, we have found that column smoothing produces better certified accuracies than block smoothing. However, the performance gap is larger on MNIST than on CIFAR-10. We have also tested with the base classifier returning only the top-one class, rather than

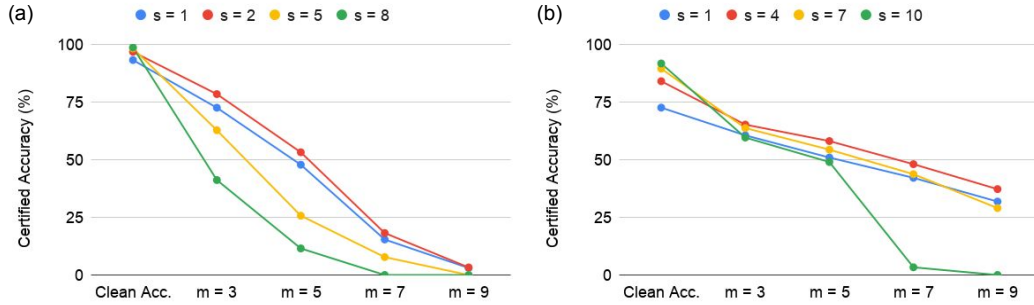

Figure 6: Validation set certificates for $m \times m$ patches on (a) MNIST, (b) CIFAR-10. For all experiments, we use Column Smoothing, $\theta = 0.3$. On CIFAR-10, we maintain high certified accuracy even at $m = 9$. The optimal column width $s$ seems not to depend on the patch size, suggesting that a single trained model will defend against a broad class of patch attacks.

thresholding the logits to abstain on low confidence predictions. We find that thresholding produces a large improvement on MNIST, but has had little effect on CIFAR-10. This is possibly because MNIST images, when ablated, will often have zero information (i.e., be entirely black), while in natural images, the retained region will always have some information. In both datasets, we found that the column smoothing certificates are not highly sensitive to the threshold hyperparameter $\theta$.

Experiments using multiple blocks and columns, rather than just a single block or column for each base classification, are presented in supplementary materials.

In Figure 6, we show how our certificates scale to different patch sizes, beyond the standard $5 \times 5$. On CIFAR-10, we maintain high certified accuracy even at a patch size of $9 \times 9$. Notably, the optimal column width $s$ seems not to depend on the patch size, suggesting that a single trained model can defend against a broad class of patch attacks.

On ImageNet-1000 (ILSVRC2012), we have tested certified robustness to $42 \times 42$ patch attacks with column smoothing alone, using column width $s = 25$, and over the $\theta$ hyperparameter range $\theta = \{0.1, 0.2, 0.3, 0.4\}$. We have used 1,000 images for validation, and 1,000 for test, using the optimal $\theta = 0.2$; test set results are presented in Table 1. Full validation results for all datasets are presented as tables in supplementary materials.

We also compare column smoothing certificates for MNIST and CIFAR-10 to *randomized* column smoothing smoothing certificates on both datasets: see Table 2. We find that the "derandomization" improves the certificates independently of the effect of thresholding (for example, it increases the certified accuracy on CIFAR-10 by nearly 7 percentage points.)

| Dataset | Derandomized $\theta = .3$ | Derandomized Top-1 class | Randomized Column Smoothing |
|---|---|---|---|
| MNIST | 53.22% (s = 2) | 22.20% (s = 6) | 16.32% (s = 6) |
| CIFAR-10 | 58.08% (s = 4) | 57.36% (s = 4) | 50.38% (s = 6) |

Table 2: Comparison of Certified Accuracies for derandomized versus randomized structured ablation (column smoothing) for $5 \times 5$ adversarial patches for MNIST and CIFAR-10. We compare randomized structured ablation to both the "Top-1 class" method (without abstaining or thesholding) as well as to the thresholding method, with the optimal $\theta = 0.3$. Here, we show the certified accuracy for the optimal value of the hyperparameter $s$ for each method: results for all $s$ are presented in supplementary materials.

## 3.1   Empirical Robustness

We evaluated the empirical robustness of our method, specifically column smoothing, on CIFAR-10, using a modified version of the IFGSM patch attack from [9]. In particular, because the zero-one base-classifications $f_c$ are non-differentiable, we cannot attack $n_c(\mathbf{x})$ directly. Instead, in order to generate the attacks, we use a surrogate model in which $f_c$ returns SoftMax scores. Note that this is similar to

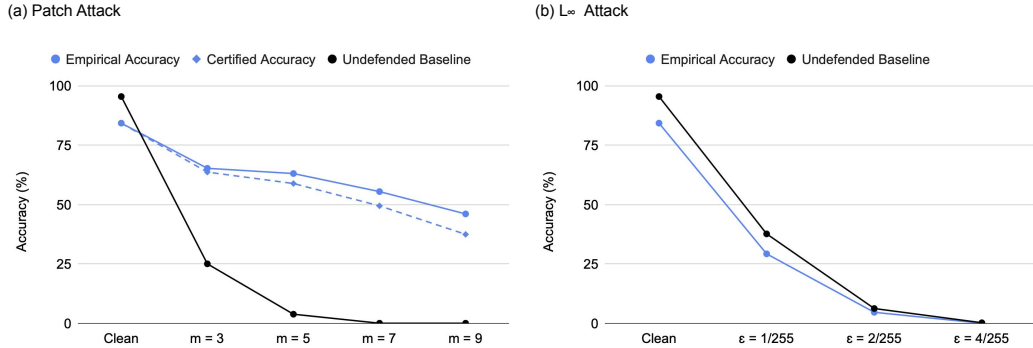

Figure 7: Empirical attacks against column smoothing on 500 images from the CIFAR-10 test set, versus an unprotected baseline model. We use optimal hyperparameters ($s = 4, \theta = 0.3$) for column smoothing. For the $L_\infty$ attack, we used IFGSM for 50 iterations and a step size of $0.5/255$. For the patch attack, we use the patch-IFGSM attack from [9], with 80 random starts, 150 iterations per random start, and step size of 0.05.

[13]'s attack on Gaussian-smoothed classifiers, but there is no need to consider random sampling in this case. Further details on the attack are provided in supplementary materials. Results are presented in Figure 7-a. We note that, as expected, our certified lower bounds hold, and furthermore that our model is significantly more robust to patch adversarial attacks compared with an undefended baseline model. We also evaluated the robustness of our attack to a non-patch adversarial attack, specifically an $L_\infty$-bounded IFGSM attack. Because all base classifiers are attacked simultaneously in this model, our method provides no robustness guarantee, and one might worry that the model could be particularly vulnerable to the attack. However, while the accuracy under this attack was reduced compared to an undefended baseline model, this was not a dramatic effect: see Figure 7-b.

## 4  Conclusion

Patch adversarial attacks are important threat models because they formalize physical adversarial attacks. In this work, we propose Structured Ablation, a provably robust defense against patch attacks. Our method, an adaptation of randomized smoothing, significantly outperforms the state-of-the-art certified defense for patch attacks on CIFAR-10, and, unlike previous methods, scales to ImageNet.

## Broader Impact

Adversarial patch attacks are extremely relevant to security threats posed by adversarial machine learning. In particular, patch attacks model physical adversarial attacks, in which real-world objects are manipulated in order to disrupt computer vision systems. Malicious use of such attacks could therefore be catastrophically damaging in highly critical applications of computer vision, such as self-driving cars. For example, consider an adversary that puts adversarial stickers on *stop signs* to cause significant errors in classification of those images by deep models deployed in autonomous vehicles. Such an attack could cause significant damage. Moreover, the existence of such vulnerabilities in deep models can harm the confidence of users of systems that employ such models, which could slow adoption of these systems. Our proposed techniques in this paper provide new provable and guaranteed defenses against these attacks, advancing the effort to mitigate these issues.

We note that the algorithms described in this paper are purely defensive. That is, this work does not reveal (in any way obvious to the authors) any unknown vulnerabilities in existing computer vision systems. While we do develop an adversarial attack against our classifier, this attack is a straightforward extension of existing work on adversarial attacks to smoothed classifiers [13]: implementing this attack represents necessary due diligence to evaluate the robustness of our defense.

One possible negative outcome of provable robustness guarantees is that they may cause users to be overconfident in the reliability of machine learning systems in general. As we have demonstrated in Section 3.1, our techniques do *not* provide robustness to general adversarial attacks other than patch

attacks. Further, a guarantee of robustness is not a guarantee of correctness: in fact, the accuracy of our classifiers is reduced compared to undefended models. Users should be aware of these issues before applying these techniques in critical applications.

Additionally, we acknowledge that, as for any computer vision advance, malicious actors could also use the techniques demonstrated here. For example, the application of these techniques could make it more difficult to thwart excessive and unwanted surveillance, posing a potential privacy concern. However, given the important safety applications described above, we believe that making this work available will have an overall positive impact.

## Acknowledgements

This project was supported in part by NSF CAREER AWARD 1942230, grants from NIST 60NANB20D134, HR001119S0026, HR00111990077, HR00112090132 and Simons Fellowship on "Foundations of Deep Learning."

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
