[Supplementary Material · Derandomized_smoothing_Neurips_Supplement.pdf]

# Supplementary Materials
# (De)Randomized Smoothing for Certifiable Defense against Patch Attacks

**Alexander Levine and Soheil Feizi**
Department of Computer Science
University of Maryland College Park, MD 20742
{alevine0, sfeizi}@cs.umd.edu

## A   Proofs

We first prove the block smoothing algorithm. Recall the definitions and statement of Theorem 1. In particular, recall the base classification counts $n_c(\mathbf{x})$:

$$\forall c, \ n_c(\mathbf{x}) := \sum_{x=1}^{w} \sum_{y=1}^{h} f_c(\mathbf{x}, s, x, y) \tag{1}$$

And recall the definition of the smoothed classifier:

$$g(\mathbf{x}) := \arg\max_c n_c(\mathbf{x}), \tag{2}$$

where in the case of ties, we choose the smaller-indexed class as the argmax solution.

**Theorem 1.** *For any image* $\mathbf{x}$*, base classifier* $f$*, smoothing block size* $s$*, and patch size* $m$*, if:*

$$n_c(\mathbf{x}) \geq \max_{c' \neq c} \left[ n_{c'}(\mathbf{x}) + \mathbf{1}_{c>c'} \right] + 2(m+s-1)^2 \tag{3}$$

*then for any image* $\mathbf{x}'$ *which differs from* $\mathbf{x}$ *only in an* $(m \times m)$ *patch,* $g(\mathbf{x}') = c$.

*Proof.* Let $(i, j)$ represent the upper-right corner of the $m \times m$ patch in which $\mathbf{x}$ and $\mathbf{x}'$ differ. Note that, for all $c$, the output of $f_c(\mathbf{x}, s, x, y)$ will be equal to the output of $f_c(\mathbf{x}', s, x, y)$, unless the $s \times s$ block retained (starting at $(x, y)$) intersects with the $m \times m$ adversarial patch (starting at $(i, j)$). This condition occurs only when both $x$ is in the range between $i - s + 1$ and $i + m - 1$, inclusive, and $y$ is in the range between $j - s + 1$ and $j + m - 1$, inclusive. Note that there are $(m + s - 1)$ values each for $x$ and $y$ which meet this condition, and therefore $(m + s - 1)^2$ such pairs $(x, y)$. Therefore $f_c(\mathbf{x}, s, x, y) = f_c(\mathbf{x}', s, x, y)$ in all but $(m + s - 1)^2$ cases.

Note that if $i - s + 1 < 0$, then the intersecting values for $x$, taking into account the wrapping behavior of $f$, will be $h - (i - s + 1)$ through $h$, and 0 through $i + m - 1$ (see Figure 4 in the main text): there are still $(m + s - 1)$ such values, and a similar argument applies to $j$.

Therefore, because $f_c(\cdot) \in \{0, 1\}$,

$$\forall c, |n_c(\mathbf{x}) - n_c(\mathbf{x}')| \leq (m+s-1)^2. \tag{4}$$

Now, consider any $c' \neq c$, such that $n_c(\mathbf{x}) \geq [n_{c'}(\mathbf{x}) + \mathbf{1}_{c>c'}] + 2(m+s-1)^2$. There are two cases:

- $c > c'$: In this case, in the event that $n_c(\mathbf{x}') = n'_c(\mathbf{x}')$, we have that $g(\mathbf{x}') = c'$. Therefore, a sufficient condition for $g(\mathbf{x}') \neq c'$ is that $n_c(\mathbf{x}') > n'_c(\mathbf{x}')$. By Equation 4 and triangle inequality, this must be true if $n_c(\mathbf{x}) > [n_{c'}(\mathbf{x})] + 2(m + s - 1)^2$, or equivalently, if $n_c(\mathbf{x}) \geq [n_{c'}(\mathbf{x}) + \mathbf{1}_{c>c'}] + 2(m + s - 1)^2$.

- $c' > c$: In this case, in the event that $n_c(\mathbf{x}') = n'_c(\mathbf{x}')$, we have that $g(\mathbf{x}') = c'$. Therefore, a sufficient condition for $g(\mathbf{x}') \neq c'$ is that $n_c(\mathbf{x}') \geq n'_c(\mathbf{x}')$. By Equation 4 and triangle inequality, this must be true if $n_c(\mathbf{x}) \geq [n_{c'}(\mathbf{x}) + \mathbf{1}_{c>c'}] + 2(m+s-1)^2$.

Therefore, if $n_c(\mathbf{x}) \geq \max_{c'\neq c}[n_{c'}(\mathbf{x}) + \mathbf{1}_{c>c'}] + 2(m+s-1)^2$, then no class other than $c$ can be output by $g(\mathbf{x}')$. $\qquad\square$

The column smoothing method can be proved similarly. For completeness, we state and prove Theorem 2 here as well. Recall

$$\forall c, \quad n_c(\mathbf{x}) := \sum_{x=1}^{w} f_c(\mathbf{x}, s, x) \tag{5}$$

**Theorem 2.** *For any image* $\mathbf{x}$*, base classifier* $f$*, smoothing band size* $s$*, and patch size* $m$*, if:*

$$n_c(\mathbf{x}) \geq \max_{c'\neq c}[n_{c'}(\mathbf{x}) + \mathbf{1}_{c>c'}] + 2(m+s-1) \tag{6}$$

*then for any image* $\mathbf{x}'$ *which differs from* $\mathbf{x}$ *only in an* $(m \times m)$ *patch,* $g(\mathbf{x}') = c$.

*Proof.* Let $(i,j)$ represent the upper-right corner of the $m \times m$ patch in which $\mathbf{x}$ and $\mathbf{x}'$ differ. Note that, for all $c$, the output of $f_c(\mathbf{x}, s, x)$ will be equal to the output of $f_c(\mathbf{x}', s, x)$, unless the band (of width $s$) retained, starting at column $x$, intersects with the $m \times m$ adversarial patch (starting at $(i,j)$). This condition occurs only when $x$ is in the range between $i-s+1$ and $i+m-1$, inclusive. Note that there are $(m+s-1)$ values for $x$ which meet this condition. Therefore $f_c(\mathbf{x}, s, x, y) = f_c(\mathbf{x}', s, x, y)$ in all but $(m+s-1)$ cases.

Again, if $i-s+1 < 0$, then the intersecting values for $x$, taking into account the wrapping behavior of $f$ will be $h-(i-s+1)$ through $h$, and $0$ through $i+m-1$: there are still $(m+s-1)$ such values. Therefore, because $f_c(\cdot) \in \{0,1\}$,

$$\forall c, |n_c(\mathbf{x}) - n_c(\mathbf{x}')| \leq (m+s-1). \tag{7}$$

The rest of the proof proceeds exactly as in the block smoothing case, with $(m+s-1)$ substituted for $(m+s-1)^2$. $\qquad\square$

## B   Full Validation Result Tables for Column and Block Smoothing

Tables 1 and 3 present the full validation set clean and certified accuracies for $5 \times 5$ patches on MNIST and CIFAR-10, respectively, for all tested values of parameters $s$ and $\theta$, and for both block and column smoothing. Note that this is presented in Figure 5 in the main text. Table 2 presents the validation set clean and certified accuracies for $42 \times 42$ patches on ImageNet using column smoothing, for all four tested values of the hyperparameter $\theta$.

## C   Results for Row Smoothing

We also tested smoothing with rows, rather than columns, on MNIST. This resulted in slightly lower certified accuracy under $5 \times 5$ patch attacks (45.32% validation set certified accuracy, versus 53.22% using column smoothing). Full results are presented in Table 4.

## D   Multi-column and Multi-block Derandomized Smoothing

In the main text, we argued for having the base classifier use a single contiguous group of pixels on the grounds that, compared to selecting individual pixels, it provides for a smaller risk of intersecting the adversarial patch. However, there may be some benefit to getting information from multiple distinct areas of an image, even if there is some associated increase in $\Delta$. Rather than just looking at the extremes of entirely independent pixels (Table 2 in the main text) versus a single band or block (Figure 5 in the main text) we also explored, on MNIST, the intermediate case of using a small number of bands or blocks. In Table 5, we show all mathematically possible multiple-column

| Column Size $s$ | $\theta = .2$ | | $\theta = .3$ | | $\theta = .4$ | | Top-1 class (no threshold) | |
|---|---|---|---|---|---|---|---|---|
| | Clean Acc | Cert Acc | Clean Acc | Cert Acc | Clean Acc | Cert Acc | Clean Acc | Cert Acc |
| 1 | 93.50% | 47.52% | 93.22% | 47.82% | 92.56% | 45.42% | 50.04% | 14.78% |
| 2 | 96.68% | 51.46% | **96.78%** | **53.22%** | 96.36% | 52.34% | 72.80% | 19.22% |
| 3 | 97.84% | 45.92% | 97.70% | 47.22% | 97.46% | 39.14% | 82.36% | 19.48% |
| 4 | 97.88% | 38.92% | 97.92% | 32.98% | 97.84% | 32.52% | 85.46% | 19.86% |
| 5 | 98.24% | 32.62% | 98.26% | 25.72% | 98.06% | 25.26% | 93.20% | 21.50% |
| 6 | 98.44% | 27.60% | 98.30% | 21.52% | 98.24% | 20.42% | 95.42% | 22.20% |
| 7 | 98.58% | 14.14% | 98.60% | 15.94% | 98.56% | 15.98% | 97.56% | 20.34% |
| 8 | 98.70% | 10.04% | 98.68% | 11.52% | 98.70% | 11.76% | 97.90% | 18.90% |
| 9 | 98.88% | 06.52% | 98.82% | 08.16% | 98.74% | 08.32% | 98.48% | 17.28% |
| **Block Size $s$** | $\theta = .2$ | | $\theta = .3$ | | $\theta = .4$ | | Top-1 class (no threshold) | |
| 1 | 09.76% | 0% | 09.76% | 0% | 09.76% | 0% | 10.80% | 10.80% |
| 2 | 09.76% | 0% | 09.76% | 0% | 09.76% | 0% | 10.80% | 10.80% |
| 3 | 09.76% | 0% | 09.76% | 0% | 09.76% | 0% | 10.80% | 10.80% |
| 4 | 87.30% | 38.06% | 86.04% | 29.62% | 85.94% | 19.32% | 12.78% | 10.80% |
| 5 | 91.60% | 42.58% | 90.24% | 36.52% | 90.32% | 27.22% | 21.72% | 11.08% |
| 6 | 93.44% | 42.90% | 92.68% | 39.86% | 92.70% | 33.06% | 31.60% | 11.74% |
| 7 | 94.78% | 44.00% | 94.30% | 41.80% | 94.52% | 35.84% | 51.18% | 13.30% |
| 8 | 96.04% | 44.04% | 95.64% | 42.22% | 95.66% | 36.66% | 75.94% | 17.64% |
| 9 | 96.96% | 41.74% | 97.02% | 41.84% | 96.92% | 37.18% | 91.74% | 26.84% |
| 10 | 97.54% | 39.84% | 97.44% | 40.00% | 97.50% | 36.02% | 95.66% | 35.30% |
| 11 | 97.88% | 36.00% | 97.66% | 36.64% | 97.64% | 32.34% | 96.58% | 31.26% |
| 12 | 98.10% | 30.40% | 98.38% | 28.26% | 98.30% | 32.98% | 96.98% | 27.26% |
| 13 | 98.38% | 28.26% | 98.30% | 29.06% | 98.44% | 22.72% | 98.02% | 27.66% |
| 14 | 98.70% | 22.22% | 98.62% | 18.68% | 98.62% | 14.54% | 98.40% | 24.04% |
| 15 | 98.86% | 08.90% | 98.84% | 08.00% | 98.86% | 06.12% | 98.68% | 12.90% |

Table 1: Validation set clean and certified accuracies for $5 \times 5$ patch adversarial attacks using Block and Column smoothing on MNIST, with results shown for all tested values of parameters $s$ and $\theta$. The value with the highest certified accuracy is shown in bold.

| | Clean Accuracy | Certified Accuracy |
|---|---|---|
| $s = 25, \ \theta = 0.1$ | 44.0% | 12.0% |
| $s = 25, \ \theta = 0.2$ | **43.1%** | **14.5%** |
| $s = 25, \ \theta = 0.3$ | 42.3% | 13.8% |
| $s = 25, \ \theta = 0.4$ | 40.9% | 12.3% |

Table 2: Validation set clean and certified accuracies for $42 \times 42$ patch adversarial attacks using Column smoothing on ImageNet, with results shown for all tested values of parameter $\theta$. The value with the highest certified accuracy is shown in bold.

certificates on MNIST, as well as several certificates for multiple-blocks with $s = 4$. Interestingly, while the certificates using multiple columns are far below optimal, the certified accuracy for two blocks is only marginally below the best single-block certified accuracy.

For smoothing with multiple blocks or multiple columns, we consider only blocks or columns aligned to a grid starting at the upper-left corner of the image. For example, if using block size $s = 4$, we consider only retaining blocks with upper-left corner $(i, j)$, where $i$ and $j$ are both multiples of 4. This prevents retained blocks from overlapping, and also reduces the (large) number of possible selections of multiple blocks, allowing for derandomized smoothing.

Let the number of retained blocks or bands be $\kappa$, and, as in the paper, let the block or band size be $s$, the image size be $h \times w$, and the adversarial patch size be $m \times m$. For the block case, note that there are $\lceil h/s \rceil \times \lceil w/s \rceil$ such axis-aligned blocks. Of these, the adversarial patch will overlap at most $(\lceil (m-1)/s \rceil + 1)^2$ blocks. For example, for a $5 \times 5$ adversarial patch, using block size

| Column Size $s$ | $\theta = .2$ | | $\theta = .3$ | | $\theta = .4$ | | Top-1 class (no threshold) | |
|---|---|---|---|---|---|---|---|---|
| | Clean Acc | Cert Acc | Clean Acc | Cert Acc | Clean Acc | Cert Acc | Clean Acc | Cert Acc |
| 1 | 72.92% | 50.74% | 72.58% | 50.94% | 72.90% | 50.32% | 72.30% | 50.94% |
| 2 | 77.54% | 53.26% | 77.70% | 54.14% | 77.68% | 53.04% | 77.10% | 53.40% |
| 3 | 81.84% | 56.14% | 81.74% | 56.76% | 81.98% | 56.08% | 81.82% | 56.24% |
| 4 | 84.04% | 56.62% | **84.04%** | **58.08%** | 84.16% | 57.12% | 83.66% | 57.36% |
| 5 | 85.98% | 55.18% | 85.66% | 56.08% | 85.82% | 55.98% | 85.32% | 56.00% |
| 6 | 87.70% | 54.84% | 87.90% | 56.26% | 88.04% | 56.10% | 87.62% | 55.70% |
| 7 | 89.24% | 53.12% | 89.48% | 54.36% | 89.30% | 54.04% | 89.12% | 54.14% |
| 8 | 90.60% | 51.38% | 90.68% | 52.90% | 90.60% | 53.12% | 90.34% | 53.02% |
| 9 | 91.38% | 47.78% | 91.30% | 49.96% | 91.38% | 50.30% | 91.12% | 50.36% |
| 10 | 91.66% | 46.26% | 91.74% | 49.00% | 91.62% | 49.44% | 91.56% | 49.58% |
| 11 | 92.40% | 41.24% | 92.26% | 45.12% | 92.18% | 46.08% | 92.18% | 45.90% |
| **Block Size $s$** | $\theta = .2$ | | $\theta = .3$ | | $\theta = .4$ | | Top-1 class (no threshold) | |
| 1 | 14.94% | 12.44% | 14.70% | 12.42% | 13.62% | 10.64% | 14.82% | 12.80% |
| 2 | 29.94% | 20.96% | 25.30% | 17.06% | 22.66% | 14.00% | 29.48% | 22.58% |
| 3 | 41.88% | 27.70% | 36.34% | 24.24% | 32.88% | 18.14% | 39.52% | 27.80% |
| 4 | 27.88% | 18.80% | 31.10% | 18.68% | 31.98% | 16.90% | 28.12% | 18.34% |
| 5 | 58.64% | 37.72% | 57.58% | 35.74% | 56.00% | 29.20% | 56.98% | 39.64% |
| 6 | 68.86% | 45.88% | 67.70% | 44.46% | 66.26% | 39.00% | 67.52% | 46.20% |
| 7 | 71.98% | 46.96% | 72.02% | 47.40% | 71.32% | 43.02% | 71.26% | 48.38% |
| 8 | 74.90% | 49.18% | 74.80% | 49.96% | 75.78% | 46.72% | 74.24% | 50.68% |
| 9 | 79.18% | 52.04% | 78.82% | 53.04% | 79.50% | 50.28% | 78.42% | 53.88% |
| 10 | 82.32% | 53.44% | 82.56% | 54.82% | 82.96% | 52.50% | 82.00% | 55.18% |
| 11 | 84.34% | 52.84% | 84.94% | 54.94% | 85.12% | 52.84% | 84.40% | 55.24% |
| 12 | 86.56% | 53.26% | 86.66% | 55.66% | 86.88% | 54.34% | 86.38% | 56.08% |
| 13 | 88.50% | 51.76% | 88.40% | 54.26% | 88.98% | 53.52% | 88.28% | 54.74% |
| 14 | 89.98% | 50.72% | 89.86% | 53.94% | 90.22% | 53.22% | 89.86% | 54.58% |
| 15 | 90.94% | 49.88% | 91.12% | 52.70% | 91.28% | 52.70% | 90.92% | 53.44% |
| 16 | 92.04% | 46.04% | 91.98% | 49.26% | 92.02% | 49.46% | 91.84% | 50.12% |
| 17 | 91.82% | 40.30% | 92.04% | 44.14% | 92.12% | 44.74% | 92.12% | 45.14% |
| 18 | 93.42% | 28.42% | 93.52% | 32.46% | 93.54% | 33.36% | 93.44% | 33.98% |

Table 3: Validation set clean and certified accuracies for $5 \times 5$ patch adversarial attacks using Block and Column smoothing on CIFAR-10, with results shown for all tested values of parameters $s$ and $\theta$. The value with the highest certified accuracy is shown in bold.

$s = 4$, the adversarial patch will overlap exactly 4 blocks, regardless of position: see Figure 1. When performing derandomized smoothing, we classify all $\binom{\lceil h/s \rceil \times \lceil w/s \rceil}{\kappa}$ possible choices of $\kappa$ blocks. Of these classifications, at least

$$\binom{\lceil \frac{h}{s} \rceil \times \lceil \frac{w}{s} \rceil - (\lceil \frac{m-1}{s} \rceil + 1)^2}{\kappa}$$

will use none of the at most $(\lceil (m-1)/s \rceil + 1)^2$ blocks which may be affected by the adversary. Therefore, the number of classifications which might be affected by the adversary is at most:

$$\binom{\lceil \frac{h}{s} \rceil \times \lceil \frac{w}{s} \rceil}{\kappa} - \binom{\lceil \frac{h}{s} \rceil \times \lceil \frac{w}{s} \rceil - (\lceil \frac{m-1}{s} \rceil + 1)^2}{\kappa}.$$

We can then use the above quantity in place of the number of classifications $(m + s - 1)^2$ that might be affected by the adversarial patch in standard block smoothing (Equation 4). This modification, in addition to classifying all $\binom{\lceil h/s \rceil \times \lceil w/s \rceil}{\kappa}$ selections of $\kappa$ axis-aligned blocks, is sufficient to adapt the certification algorithm to a multi-block setting.

The column case is similar: there are $\lceil w/s \rceil$ axis-aligned bands (defined as bands which start at a column index which is a multiple of $s$). Of these, the adversarial patch will overlap at most

| Row Size $s$ | $\theta = .2$ | | $\theta = .3$ | | $\theta = .4$ | |
|---|---|---|---|---|---|---|
| | Clean Accuracy | Certified Accuracy | Clean Accuracy | Certified Accuracy | Clean Accuracy | Certified Accuracy |
| 1 | 88.46% | 36.54% | 85.26% | 33.78% | 82.86% | 25.52% |
| 2 | 95.58% | 43.52% | **93.92%** | **45.32%** | 92.04% | 43.16% |
| 3 | 96.28% | 41.80% | 95.26% | 44.96% | 94.08% | 43.74% |
| 4 | 97.26% | 38.58% | 96.40% | 42.02% | 95.70% | 41.82% |
| 5 | 97.74% | 35.74% | 97.00% | 39.04% | 96.52% | 39.54% |
| 6 | 97.60% | 32.18% | 97.18% | 36.98% | 96.92% | 37.10% |
| 7 | 98.04% | 27.32% | 97.62% | 32.82% | 97.48% | 33.50% |
| 8 | 98.30% | 23.16% | 98.18% | 28.26% | 98.06% | 29.54% |
| 9 | 98.24% | 17.60% | 97.96% | 23.80% | 97.92% | 25.12% |

Table 4: Validation set clean and certified accuracies for $5 \times 5$ patch adversarial attacks using Row smoothing on MNIST. Values with highest certified accuracies are shown in bold.

| | $\theta = .2$ | | $\theta = .3$ | | $\theta = .4$ | |
|---|---|---|---|---|---|---|
| | Clean Accuracy | Certified Accuracy | Clean Accuracy | Certified Accuracy | Clean Accuracy | Certified Accuracy |
| 2 columns, $s = 1$ | 96.80% | 38.12% | **96.50%** | **39.18%** | 96.24% | 37.38% |
| 2 columns, $s = 2$ | **98.38%** | **31.36%** | 98.24% | 25.56% | 98.10% | 25.80% |
| 3 columns, $s = 1$ | 97.74% | 07.58% | **97.68%** | **09.36%** | 97.64% | 09.00% |
| 2 blocks, $s = 4$ | **92.32%** | **43.40%** | 91.22% | 38.78% | 91.32% | 30.08% |
| 3 blocks, $s = 4$ | **94.98%** | **41.40%** | 94.42% | 39.38% | 94.46% | 32.62% |
| 4 blocks, $s = 4$ | **96.26%** | **38.26%** | 95.72% | 37.50% | 95.72% | 32.02% |

Table 5: Multi-column and multi-block certificates, with results shown for all tested values of parameter $\theta$. Results are on the MNIST validation set, for $5 \times 5$ patches. For each number of blocks/columns and block/column size $s$, we bold the highest certified accuracy over tested values of the hyperparameter $\theta$.

$(\lceil (m-1)/s \rceil + 1)$ bands. When performing smoothing, we classify all $\binom{\lceil w/s \rceil}{\kappa}$ possible choices of $\kappa$ bands. Of these classifications, at least

$$\binom{\lceil \frac{w}{s} \rceil - (\lceil \frac{m-1}{s} \rceil + 1)}{\kappa}$$

will use none of the at most $(\lceil (m-1)/s \rceil + 1)$ bands which may be affected by the adversary. Therefore, the number of classifications which might be affected by the adversary is at most:

$$\binom{\lceil \frac{w}{s} \rceil}{\kappa} - \binom{\lceil \frac{w}{s} \rceil - (\lceil \frac{m-1}{s} \rceil + 1)}{\kappa}.$$

Full validation set results for multi-block and multi-band smoothing are shown in Table 5.

Figure 1: Multi-block smoothing: for a $5 \times 5$ adversarial patch, using block size $s = 4$, the adversarial patch overlaps exactly 4 blocks, regardless of position. Individual pixels are represented by black gridlines. Blocks that may be retained are outlined in blue, and three possible $5 \times 5$ adversarial patches are shown in red. Note that this is exact because, in this case, $m - 1$ is divisible by $s$: in other cases, some choices of adversarial patches may affect fewer than $(\lceil (m-1)/s \rceil + 1)^2$ blocks.

# E Comparison with *Randomized* Structured Ablation

As discussed in the main text, there are two benefits to derandomization: first, we can eliminate estimation error, and second, it allows the classifier to abstain or select multiple classes without complicating estimation. In order to distinguish these effects, we present in Tables 6 and 7 the certificates on MNIST and CIFAR-10 using *randomized* column smoothing (with the estimation scheme from [12]), versus deterministic column smoothing. We compare to both the "Top-1 class" method (without abstaining or thesholding) as well as to the thresholding method, with $\theta = 0.3$. We find that derandomization alone, without the thresholding method, provides a considerable improvement (around 6 percentage points increase on MNIST, around 7 percentage points on CIFAR-10). On MNIST (although not on CIFAR-10), the thresholding scheme provides a large additional improvement.

| Column Size $s$ | Derandomized $\theta = .3$ | Derandomized Top-1 class | Randomized Column Smoothing |
|---|---|---|---|
| 1 | 47.82% | 14.78% | 11.80% |
| 2 | **53.22%** | 19.22% | 14.26% |
| 3 | 47.22% | 19.48% | 15.14% |
| 4 | 32.98% | 19.86% | 15.24% |
| 5 | 25.72% | 21.50% | 15.48% |
| 6 | 21.52% | **22.20%** | **16.32%** |
| 7 | 15.94% | 20.34% | 14.50% |
| 8 | 11.52% | 18.90% | 14.52% |
| 9 | 08.16% | 17.28% | 14.10% |

Table 6: Comparison of Derandomized vs. Randomized Structured Ablation certified accuracies for $5 \times 5$ adversarial patches on MNIST.

| Column Size $s$ | Derandomized $\theta = .3$ | Derandomized Top-1 class | Randomized Column Smoothing |
|---|---|---|---|
| 1 | 50.94% | 50.94% | 38.16% |
| 2 | 54.14% | 53.40% | 41.98% |
| 3 | 56.76% | 56.24% | 47.02% |
| 4 | **58.08%** | **57.36%** | 49.56% |
| 5 | 56.08% | 56.00% | 49.58% |
| 6 | 56.26% | 55.70% | **50.38%** |
| 7 | 54.36% | 54.14% | 50.04% |
| 8 | 52.90% | 53.02% | 48.94% |
| 9 | 49.96% | 50.36% | 47.28% |
| 10 | 49.00% | 49.58% | 46.46% |
| 11 | 45.12% | 45.90% | 43.28% |

Table 7: Comparison of Derandomized vs. Randomized Structured Ablation certified accuracies for $5 \times 5$ adversarial patches on CIFAR-10.

# F Sparse Randomized Ablation for Patch adversarial Attacks

In Table 8, we provide the certified accuracies computed from applying sparse Randomized Ablation [16] to patch adversarial attacks, as discussed in Section 2.1 of the main text.

# G Adversarial Attack Details

In order to test adversarial attacks against our structured ablation model (in particular the column smoothing model) we must work around the non-differentiability of the base classifier $f$ with respect to the image. We accomplish this using a method similar to the attack on smooth classifiers proposed by [13].

| MNIST | | | CIFAR-10 | | |
|---|---|---|---|---|---|
| Retained pixels $k$ | Classification accuracy | Certified accuracy | Retained pixels $k$ | Classification accuracy | Certified accuracy |
| 5 | 32.44% | 7.58% | 25 | 68.28% | 13.28% |
| 10 | 75.02% | 5.40% | 50 | 74.68% | 0 |
| 15 | 86.32% | 4.34% | 75 | 78.26% | 0 |
| 20 | 90.36% | 0.10% | 100 | 80.98% | 0 |
| 25 | 93.20% | 0 | 125 | 83.82% | 0 |
| 30 | 94.72% | 0 | 150 | 85.70% | 0 |

Table 8: Certified accuracy to $5 \times 5$ adversarial patches from directly applying $L_0$ smoothing as proposed by [16]. Note that with $L_0$ smoothing, the geometry of the attack is not taken into consideration: these are therefore actually certified accuracies for any $L_0$ attack on up to $\rho = 25$ pixels. The certificates are probabilistic, with $95\%$ confidence.

In particular, as described in Section 2.2.1 in the main text, the base classifier $f$ in our model is implemented using a neural network: let $F$ represent the (SoftMax-ed) logits of this neural network:

$$f_c(\mathbf{x}, s, x) = \begin{cases} 1, & \text{if } F_c(\mathbf{x}, s, x) \geq \theta \\ 0, & \text{if } F_c(\mathbf{x}, s, x) < \theta \end{cases} \tag{8}$$

Rather than attacking $n(\mathbf{x}) = \sum_{x=1}^{w} f(\mathbf{x}, s, x)$, we instead attack a soft smooth classifier, $N(\mathbf{x})$:

$$N_c(\mathbf{x}) := \frac{1}{w} \sum_{x=1}^{w} F_c(\mathbf{x}, s, x) \tag{9}$$

The objective of the adversary (as in [9]) is now applied to this soft classifier:

$$\max_{\mathbf{x} \in \text{ (Patch Constraints)}} -\log\left(\frac{1}{w} \sum_{x=1}^{w} F_y(\mathbf{x}, s, x)\right) \tag{10}$$

where $y$ is the true label. The IFGSM patch attack proposed by [9] proceeds by first randomly selecting a patch to attack, and then attacking it with standard IFGSM, without imposing any $L_\infty$ magnitude constraint on the attack (other than as required to produce a feasable image). This is repeated many times on many random patches. However, the most successful attack so far is recorded at each step of optimization, and finally returned at the end of the attack. (Note that this is the most successful attack over all steps of all random initializations.) In [9], this is taken as whichever perturbed version of the image maximizes the objective (Equation 10). Because we ultimately care about the "hard" smoothed classifier $n(\mathbf{x})$, we instead just evaluate the final "hard" classification $n(\mathbf{x})$ at each step. We record an attack to return only if it is actually successful at making the final classification incorrect. Note that this does not impose significant computational costs, because we already have the value of each 'soft' base classifier $F_c((\mathbf{x})$ at each step.

As mentioned in the main text, for the patch attack, we perform 80 random starts, 150 iterations per random start, and use a step size of 0.05. When attacking patches, we uniformly randomly initialize the pixels in the attacked region. For $L_\infty$ IFGSM, we used IFGSM for 50 iterations and a step size of $0.5/255$: for this, we did not randomize the pixel values before optimizing, but rather started at the initial $\mathbf{x}$. Training parameters for baseline models were identical to those for column-smoothed models, except that a regular, full ResNet-18 model was used.

## H  Evaluation Times

Data on evaluation times (using the optimal hyperparameters to maximize certified accureacy for each method) are shown in Table 9. We used NVIDIA 2080 Ti GPUs for our experiments.

## I  Architecture and Training Details

As discussed in the paper, we used the method introduced by [16] to represent images with pixels ablated: this requires increasing the number of input channels from one to two for greyscale images

| Method and Dataset | Images | Seconds | GPUs | GPU-seconds/image |
|---|---|---|---|---|
| Column, MNIST | 5000 | 6.61 | 1 | 0.00132 |
| Block, MNIST | 5000 | 48.1 | 1 | 0.00962 |
| Column, CIFAR-10 | 5000 | 30.8 | 1 | 0.00616 |
| Block, CIFAR-10 | 5000 | 851 | 1 | 0.170 |
| Column, ImageNet | 1000 | 622 | 4 | 2.49 |

Table 9: Evaluation times. Note that evaluation and certification both require evaluating each base classifier, so these are also the certification times (our evaluation script reports both clean and certified accuracy).

| | MNIST | CIFAR-10 | ImageNet |
|---|---|---|---|
| Training Epochs | 400 | 350 | 60 |
| Batch Size | 128 | 128 | 196 |
| Training Set Preprocessing | None | Random Cropping (Padding:4) and Random Horizontal Flip | Random Horizontal Flip |
| Optimizer | Stochastic Gradient Descent with Momentum | Stochastic Gradient Descent with Momentum | Stochastic Gradient Descent with Momentum |
| Learning Rate | .01 (Epochs 1-200) .001 (Epochs 201-400) | .1 (Epochs 1-150) .01 (Epochs 151-250) .001 (Epochs 251-350) | .1 (Epochs 1-20) .01 (Epochs 21-40) .001 (Epochs 41-60) |
| Momentum | 0.9 | 0.9 | 0.9 |
| $L_2$ Weight Penalty | 0.0005 | 0.0005 | 0.0005 |

Table 10: Training Parameters

(MNIST) and from three to six for color images. For MNIST, we used the simple CNN architecture from the released code of [16], consisting of two convolutional layers and three fully-connected layers. For CIFAR-10 and ImageNet, we used modified versions ResNet-18 and ResNet-50, respectively, with the number of input channels increased to six. Training details are presented in Table 10.

For randomized smoothing experiments, we follow the empirical estimation methods proposed by [12]. We certify to $95\%$ confidence, using 1000 random samples to select the putative top class, and 10000 random samples to lower-bound the probability of this class. For sparse randomized ablation on MNIST, we use released pretrained models from [16].