[Reviews · NeurIPS 2020]

Review 1

Summary and Contributions: The paper proposes a small modification to an existing defense to patch attacks by structuring the ablations instead of randomly sampling individual pixels.

Strengths: The new defense achieves state of the art performance, and there's decent empirical evaluation.

Weaknesses: First of all, I don't see any technical contribution; both theorems are simple counting arguments, and the (de)randomization is not really a contribution but more of a side-effect since we can just sample every possible structured ablation. Am I correct in my understanding that a different model has to be trained for each setting of s? This is sort of an issue because based on figure 5 it looks like one step too far and the accuracy tanks. I appreciate the thoroughness of the paper, and I think it's well constructed. But ultimately, this work just feels like a marginal contribution on top of [1]. I don't see enough of an idea formed in this work to merit an entire conference paper, especially with no technical contribution. [1] Robustness certificates for sparse adversarial attacks by randomized ablation. Levine and Feizi 2020.

Correctness: Yes, it's just a counting argument on top of randomized smoothing.

Clarity: Yes.

Relation to Prior Work: Yes

Reproducibility: Yes

Additional Feedback: I have read and appreciate the rebuttal. In particular, the connection between the hyperparameters s and sigma is clear, and seems like a reasonable tradeoff---but there should be a discussion of this in the paper if it is not present already. However, my main concern with this work still stands: yes, it achieves state-of-the-art performance and is the first to scale to ImageNet, but both of these facts are simply due to appropriating a *very* similar defense for a slightly different problem in an uninteresting way. This paper presents a slight variation of a slight variation of a defense. The empirical performance is great, but I feel this type of work just doesn't offer enough novelty to merit acceptance at the most competitive conferences.


Review 2

Summary and Contributions: The paper proposes a variant of randomized smoothing that is applicable for defending against patch attacks on image classifiers. For this, all but a small part of the input (either small block or narrow bands) are ablated randomly for every sample in randomized smoothing. Since there are only relatively few ways of choosing blocks/bands, this can be done even exhaustively, leading to a deterministic certificate. The main merits of the paper are that (a) the authors address an important problem in terms of safety of ML perception, namely certified robustness against physically realizable attacks in the form of patches. (b) the authors extend their method such that it provides deterministic guarantees compared to the probabilistic ones of standard randomized smoothing. (c) The deterministic variant allows adding a thresholding mechanism on the logits, which considerably improves certified accuracy. Experiments show considerable improvements of the proposed method over baselines.

Strengths: Novelty: Novel contributions are a structured ablation technique (extending Levine & Feizi, 2019) and a deterministic, derandomized smoothing variant. Experimental results show a considerable improvement over (Levine & Feizi, 2019) because of these novel components. Significance: Certifiable robustness to adversarial patches is an important goal for making perception systems robust against physical-world attacks and the paper makes considerable progress in this direction in that they increase certified robustness and clean accuracy compared to Chiang et al. [9] and Levine & Feizi [16]. Soundness of claims: proofs and empirical evaluation support the main claims of the paper (if supplementary material is taken into account) Relevance to the NeurIPS community. Adversarial attacks (particular in physically realizable threat models) and certified defenses such as the proposed work are very relevant to the NeurIPS community.

Weaknesses: Novelty:The two new contributions (structured ablation, derandomization) are relatively straight-forward/incremental. I think this is borderline novelty for NeurIPS. However, because of the strong empirical results, I lean towards considering the novelty sufficient. Significance: While the results are encouraging, several questions remain unaddressed: * how efficient is the proposed method? It requires multiple forward passes per image (in particular with block smoothing in the order of hundreds). I think these numbers (#forward passes) should be added to the main results. * given the gap between clean and certified accuracy: how would an empirical patch attack perform? that is: the paper provides a lower bound on certified robustness but the actual robustness can in principle be anywhere up to clean accuracy. An empirical attack could provide a tighter upper bound. I am aware that the thresholding with \theta makes a direct gradient-based attack infeasible but at least gradient-free attacks should be applicable. * how is robustness against whole image perturbations or common image corruptions such as ImageNet-C affected by the structured ablation approach? Soundness of claims: the results supporting the second contribution (derandomization) are only in the supplementary material (part E). I think these should clearly be part of the main paper.

Correctness: I haven't found any flaws in the method.

Clarity: I think the paper is written to much in a spirit of selling the idea to the reviewers, focusing on conveying high level ideas via illustrations and teaser images: * The introduction is close to 3 pages long, including 2 tables and 1 figure which compare the proposed work to the most important baseline and a discussion of these results. While I see a benefit of "teasering" the interest of the reader, this work goes far beyond what is common in this regard. * Result and conclusion section are very short. I recommend moving some of the additional results from the supplement into the main paper (such as the comparison between derandomized and randomized structured ablation) and instead shorten the Introduction and condense Figure 2, 3, and 4. In general, the organization of the paper could be largely improved by sticking to a more standard way of structuring the paper: introduction - related work - novel methods - experimental results - conclusion and then sticking to this structure. Moreover, every contribution would ideally be evaluated separately in subsections of the experimental results.

Relation to Prior Work: Relation to prior work (Chiang et al. [9], Levine & Feizi [16]) is discussed in sufficient detail and also compared to in the experimental results.

Reproducibility: Yes

Additional Feedback: Additional points: * the threshold \theta is hidden in the section "Implementation Details" even though it plays a crucial role in the method (as evidenced by the results in Table 5). This parameter should be discussed more prominently. * Table 5 is difficult to parse because it has both a legend (containing information regarding \theta and the smoothing method) as well as line annotations (giving the parameter s). * Regarding ethical concerns: Making a system more robust can actually have also negative ethical and societal implications, For instance if the system is used for surveillance or for autonomous drones, attacks can be well justified (staying anonymous, being invisible to an attack drone). A more robust system would make this harder. ## Post-rebuttal I have read the rebuttal. I appreciate the extra effort and think that when taking all suggestions into account for the final version (as promised by the authors), this would be a nice paper for NeurIPS. I have increased my score to 7.


Review 3

Summary and Contributions: This paper proposes a certified defense against "patch" adversarial attacks, in which the adversary can make any change they want to a patch of the input image. The proposal is to first train a base classifier which can classify images based on only information from a single block of pixels; then you define a smooth classifier as the 'majority vote' prediction of the base classifier over all possible image blocks, weighted uniformly. The number of blocks is small enough that you can compute this majority vote prediction deterministically -- you don't have to use sampling as in other randomized smoothing approaches. The key to the certificate is that a change to a patch can only impact a small number of blocks, at most. For example, to use made-up numbers, suppose that a change to a [k x k] patch can only impact 4 blocks at most. Then so long as the majority-vote class had a "block count" which exceeded the closest competing class by a margin of at least 8 blocks, the prediction is provably invariant under any manipulation to a [k x k] patch. (In the worst case, the adversary would flip 4 blocks from the top class to the second class.). There is also a version which does the same thing with rows/columns rather than blocks.

Strengths: The method outperforms the only other certified defense against patch attacks.

Weaknesses: The method is extremely simple, mathematically, though that is perhaps a plus rather than a minus.

Correctness: It seems so.

Clarity: The paper is very well-written.

Relation to Prior Work: Yes, related work is covered thoroughly.

Reproducibility: Yes

Additional Feedback: ==== post-rebuttal update ==== I share R1's concern that the novelty is a little limited for NeurIPS, and I agree that the paper is close to the accept/reject border. However, I agree with R2 that the paper is probably above the bar.


Review 4

Summary and Contributions: This paper proposed a structured ablation scheme, where instead of randomly selecting pixels to use for classification, the authors select pixels in a structured form (block or band) to reduce the probability that the adversarial path is sampled.

Strengths: This paper provides detailed theoretical proof and empirical evaluation for the proposed method.

Weaknesses: Some concerns: 1. There is too much-overleaped information in Table 1, Table 2, and Figure 1. Figure 1 includes all information presented in Tables 1 and 2. 2. What’s the logic between the proposed method and [9] and [16]? Why the authors compare the proposed method with [9] first, then [16]? Why the authors only compare the computational cost with [9], but [16]? Is the computational cost a big contribution to this paper? Is that a big issue in a practical scenario? That part is weird to me and there is no further discussion about it in the rest of this paper. 3. Why the proposed column smoothing method produces better result compare with block smoothing method? 4. The accuracy drop for the Imagenet dataset is a concern, which makes the proposed method in-practical.

Correctness: Claims and methods are correct.

Clarity: Yes

Relation to Prior Work: Yes

Reproducibility: Yes

Additional Feedback: Thanks for the authors responses. I agree with R1. The empirical results are good, but I stand by my original assessment. I have decided to maintain the current score.

[Author Response · NeurIPS 2020]

**Technical novelty:** we acknowledge R#1 and R#2's concern that our method is technically relatively simple. However, despite the simplicity of our technique, we achieve performance that is **state of the art (by a large margin) for certified robustness to patch adversarial attacks on CIFAR-10**, as well as **the first scheme for certified robustness to patch adversarial attacks that scales to ImageNet.** We believe that, due to this improved performance, our algorithm warrants consideration: as R#3 states, the simplicity of our algorithm "is perhaps a plus rather than a minus." These significant empirical gains are results of our contributions in this paper including (i) the proposal of the structured ablation methods, and (ii) the proposal of de-randomization in patch smoothing.

**Inference/certification time (R#2):** In our column smoothing method, which empirically performs best, the number of forward passes required equals the width in pixels of the image (28 for MNIST, 32 for CIFAR, 299 for ImageNet). This improves from randomized smoothing methods, which typically use $\geq 10{,}000$ forward passes per image. Empirically, with one GPU, we certify the entire 10,000 image CIFAR test set in less than 42 seconds, and the entire MNIST training set in less than 11 seconds. On 4 GPUs, certifying ImageNet images averages less than one second per image, amortized. We will add exact times to the revised draft.

**Empirical Robustness to Black-Box attacks:** R#2 asks for evaluation against empirical gradient-free patch attacks. Upon the reviewer's suggestion, we used the ARMORY Adversarial Robustness Evaluation Test Bed, and tested using the Universal Patch scenario. In this scenario, the attacks are generated independently of the attacked model (blackbox attack). The dataset tested is RESISC-45 (aerial photograph classification, with 45 classes and $224 \times 224$ image resolution: in other words, the image scale is similar to ImageNet, but the number of classes is considerably fewer). The attacked patches vary in size, and cover at most 25% of the area of the image. Using the hyperparameters borrowed from ImageNet from the paper (column smoothing, $s = 25, \theta = 0.2$), our model achieved 66% accuracy under attack and 80% clean accuracy, versus 23% accuracy under attack and 93% clean accuracy for the reference, undefended classifer provided for the scenario. This shows that our model substantially reduces the effectiveness of the attack, even with no hyperparameter optimization. Our model also had 52% *certified* accuracy against $42 \times 42$ adversarial patches. We will include these results, as well as experiments on non-patch adversarial distortions.

**Dependence on block size (R#1):** The block size ($s$) in our defense controls the amount of information the base classifier can use, and so is directly analogous to (the inverse of) the smoothing standard deviation $\sigma$ in standard Gaussian randomized smoothing (Cohen et al. 2019). Like in that work (and indeed in all randomized-smoothing techniques that we are aware of), there is a tradeoff between robustness and accuracy, with accuracy falling at high variance (low $s$), and robustness falling at low variance (high $s$). The fact that $s$ must be the same at training and test time is also true of $\sigma$ in standard randomized smoothing.

**Organization Suggestions:** We will take the reviewers' suggestions, to move content out of the paper's introduction, and to incorporate content from Appendix E into the main text to better highlight the effect of derandomization. We will also improve the presentation/readability of the data in Figure 5, and remove the redundant presentation of information in Tables 1, 2 and Figure 1.

**Role of $\theta$ parameter (R#2):** As we mention, the hyperparameter $\theta$ and the thresholding scheme only have a significant effect on MNIST, not on the more realistic CIFAR-10 dataset. We will explain this more clearly in the revised draft.

**Ethical aspects of robustness (R#2)** We will add further discussions about ethical concerns in robust ML to the paper.

**Low accuracy on ImageNet (R#4):** We acknowledge that our clean accuracy on ImageNet is relatively low (44.6%). However, given that ours is the *first* work to report patch-attack certificates at ImageNet scale, we still believe that our result on ImageNet has merit. Also, we remind the reviewers that ImageNet is a 1000-class classification problem, so 44.6% top-1 accuracy (and 13.9% certified accuracy) are still highly nontrivial. For further results on a dataset with complexity intermediate between CIFAR-10 and ImageNet, see results on RESISC-45 ("Empirical Robustness" above).

**Why does Column Smoothing outperform Block Smoothing? (R#4)** As shown in Figure 3, even when the risk of sampling the adversarial patch is higher for column smoothing and the number of retained pixels is lower, column smoothing still outperforms block smoothing because the accuracy of the base classifier is higher (despite using fewer pixels). We can speculate that this is because the base classifier has access to more varied parts of the image when sampling a column, as compared to sampling a block.

**Comparisons to Chiang et al. (2020) vs Levine and Feizi (2020) (R#4, R#1)** We chose to first highlight the comparison to Chiang et al. (2020), because Chiang et al. (2020) presents a specific certified defense against patch attacks. By contrast, Levine and Feizi (2020)'s defense against *sparse* adversarial attacks only incidentally provides certificates against patch attacks: these certificates are not competitive at all for the patch threat model. The reason that we provide this comparison is because our *techniques* are related to Levine and Feizi (2020): it therefore makes more sense to provide the comparison when discussing our techniques (to show that structured ablation has a benefit over sparse randomized ablation), rather than as a "top-line" result.

[Meta-Review · NeurIPS 2020]

This paper introduces a method to defend against adversarial patch attack with a certified approach inspired by randomized smoothing. The reviewers all agreed the method was simple and provided good robustness, but there was disagreement on whether or not the proposal was sufficiently interesting or just constituted a bag of tricks. I agree with the reviewers that the idea is not technically deep, but the results are strong and a simple approach that achieves strong results is worth publishing. The ideas in this paper should apply to other areas as well.